# Three-Dimensional UAV Path Planning Based on Multi-Strategy Integrated Artificial Protozoa Optimizer

**DOI:** 10.3390/biomimetics10040201

**Published:** 2025-03-25

**Authors:** Qingbin Sun, Xitai Na, Zhihui Feng, Shiji Hai, Jinshuo Shi

**Affiliations:** School of Electronic Information Engineering, Inner Mongolia University, Hohhot 010010, China; sunqingbin@mail.imu.edu.cn (Q.S.); fengzhihui@mail.imu.edu.cn (Z.F.); haishiji@mail.imu.edu.cn (S.H.); 32356147@mail.imu.edu.cn (J.S.)

**Keywords:** UAV path planning, multi-strategy integrated artificial protozoa optimizer, tent map, refractive opposition-based learning (ROBL), dynamic optimal leadership mechanism, Cauchy mutation strategy, simulated annealing algorithm

## Abstract

Three-dimensional UAV path planning is crucial in practical applications. However, existing metaheuristic algorithms often suffer from slow convergence and susceptibility to becoming trapped in local optima. To address these limitations, this paper proposes a multi-strategy integrated artificial protozoa optimization (IAPO) algorithm for UAV 3D path planning. First, the tent map and refractive opposition-based learning (ROBL) are employed to enhance the diversity and quality of the initial population. Second, in the algorithm’s autotrophic foraging stage, we design a dynamic optimal leadership mechanism, which accelerates the convergence speed while ensuring robust exploration capability. Additionally, during the reproduction phase of the algorithm, we update positions using a Cauchy mutation strategy. Thanks to the heavy-tailed nature of the Cauchy distribution, the algorithm is less likely to become trapped in local optima during exploration, thereby increasing the probability of finding the global optimum. Finally, we incorporate the simulated annealing algorithm into the heterotrophic foraging and reproduction stages, effectively preventing the algorithm from getting trapped in local optima and reducing the impact of inferior solutions on the convergence efficiency. The proposed algorithm is validated through comparative experiments using 12 benchmark functions from the 2022 IEEE Congress on Evolutionary Computation (CEC), outperforming nine common algorithms in terms of convergence speed and optimization accuracy. The experimental results also demonstrate IAPO’s superior performance in generating collision-free and energy-efficient UAV paths across diverse 3D environments.

## 1. Introduction

The rapid development of UAVs over the past decade has transformed numerous industries, gradually becoming a key technological pillar in fields such as industry, agriculture, logistics, and environmental monitoring. Compared to traditional manned aerial vehicles, UAVs offer distinct advantages, including a compact size, operational flexibility, high mobility, and low costs, which make them highly effective for mission execution. In the industrial sector [1,2], UAVs have been widely employed for detection, monitoring, and data collection tasks, significantly enhancing operational efficiency and reducing human risks. In agriculture [3,4], UAVs equipped with sensors and spraying devices enhance farmland monitoring and pesticide application, significantly improving production accuracy. In logistics [5], UAV-based delivery systems serve as key enablers, improving operational efficiency and reducing costs. Additionally, UAVs play vital roles in urban planning [6], environmental monitoring [7], and disaster response [8], enabling rapid data acquisition, pollution tracking, and search-and-rescue operations in challenging environments. A cornerstone for these diverse applications is the essential technology of UAV path planning.

Path planning is essential for UAVs to perform complex missions. The primary goal is to generate a flight path that meets mission requirements between a specified starting point and target location, while adhering to constraints such as obstacle avoidance, safety, and efficiency. As UAV applications expand in complex environments, the significance of path planning continues to grow. An efficient path planning algorithm not only needs to identify the optimal path within a given scenario but also must be capable of rapidly responding and adapting to obstacles and sudden changes in the environment. This problem is inherently a multi-objective optimization challenge, adding significant complexity and difficulty. Traditional algorithms, such as A* [9] and Dijkstra’s [10], guarantee path optimality but often lack efficiency, particularly in high-dimensional or dense environments with limited computational resources. In contrast, random-sampling-based methods, like RRT [11], perform better in dynamic and high-dimensional spaces but may sacrifice path quality. To overcome the limitations of traditional methods, bio-inspired heuristic algorithms have become increasingly popular in UAV path planning. These algorithms simulate biological behaviors, physical phenomena, or natural evolutionary processes, providing strong global search capabilities and adaptability. The typical heuristic algorithms include particle swarm optimization (PSO) [12,13,14], ant colony optimization (ACO) [15,16], the genetic algorithm (GA) [17,18,19], grey wolf optimization (GWO) [20,21], and artificial bee colony (ABC) [22,23], which have demonstrated strong optimization capabilities in UAV mission planning, particularly in dynamic obstacle avoidance and environmental adaptability. Despite their successes, these heuristic algorithms still face challenges such as the local optimum dilemma, slow convergence, and a limited ability to balance multiple objectives effectively.

To tackle challenges in UAV path planning with bio-heuristic algorithms, researchers have proposed multiple solutions. Zhang et al. [24] incorporated the ROBL strategy into the artificial fish swarming algorithm, enhancing the initial population diversity by addressing uneven swarm distributions. Tian et al. [25] used chaotic mapping to generate uniformly distributed particles, improving the initial population quality. Both approaches enhance the global search capability by optimizing the population distribution, leading to better UAV path planning in complex environments. Haghighi et al. [26] combined particle swarm optimization with genetic algorithms, boosting the global search capability and convergence efficiency. Yu et al. [13] improved the particle swarm algorithm by integrating simulated annealing, refining global solution updates while preventing local convergence traps. Tu et al. [27] integrated the grey wolf optimizer with the Harris hawk optimizer, leveraging Harris hawks’ flight abilities and broad vision to refine grey wolf position updates. A greedy strategy further optimized position selection, accelerating convergence. Zhou et al. [28] fused the artificial bee colony algorithm with the bat algorithm, incorporating variance factors and refining intermediate results to boost local search capability. Li et al. [29] employed adaptive coefficients based on population size to steer infeasible paths toward feasible regions in firefly algorithms, improving the stability, convergence speed, and execution efficiency. There are various approaches to improving heuristic algorithms, which can involve enhancements in population initialization as well as in the local search, global search, convergence speed, and other aspects; these methods have inspired our research.

Recent research has shown that the artificial protozoa optimizer (APO) [30] successfully addresses the limitations of traditional heuristic algorithms, particularly regarding convergence to local optima. This advantage arises from its use of two dynamic scaling factors: the autotrophic–heterotrophic scaling factor and the dormant–reproduction scaling factor, which adaptively adjust the balance between exploitation and exploration. The APO algorithm optimizes problems by simulating the foraging, dormancy, and reproduction behaviors of primitive organisms, combined with population collaboration mechanisms and random search strategies, making it a highly promising tool for UAV path planning. However, in practical applications, we have observed that the APO algorithm can be time-consuming and slow to converge when searching for the optimal solution. Inspired by the aforementioned optimization methods used in other bio-inspired algorithms, we propose the IAPO algorithm, which leverages a multi-strategy fusion approach to improve the efficiency and convergence speed in finding the global optimum.

This paper introduces an improved version of the algorithm, IAPO, which integrates multiple strategies. The main contributions of this work are as follows:A population initialization method using the tent map and ROBL is employed, which promotes a uniform distribution of the population, significantly enhances the diversity of the initial population, and thereby improves the algorithm’s exploration capability.During the autotrophic foraging phase, a dynamic optimal leadership mechanism is applied by introducing leaders and a nonlinear dynamic adjustment factor, which guides the search process toward the optimal solution, thus accelerating convergence while ensuring robust exploration.In the reproduction phase, a Cauchy mutation strategy is utilized to update positions; owing to the heavy-tailed nature of the Cauchy distribution, the algorithm is more likely to escape local optima during exploration, thereby increasing the probability of finding the global optimum.In the heterotrophic foraging and dormancy phases, a simulated annealing algorithm is incorporated, allowing the algorithm to accept inferior solutions during the early stages. This not only provides a probabilistic means to escape local optima and expand the search scope but also further enhances convergence efficiency.The performance of the IAPO algorithm is validated using the CEC2022 benchmark functions, with comparative experiments conducted against nine other commonly used algorithms. The experimental results confirm the superiority of the IAPO algorithm.The application of IAPO to UAV three-dimensional path planning problems, in comparison with other algorithms, demonstrates its adaptability and reliability in UAV 3D path planning scenarios.

The structure of this paper is as follows: Section 2 introduces the standard APO algorithm, providing an overview of its foundational principles. Section 3 discusses the methodology used to improve the APO algorithm, presenting optimization results for the IAPO algorithm and other algorithms across 12 benchmark functions from CEC2022, exploration and exploitation analyses, engineering experiments, and ablation experiments. Section 4 focuses on the mathematical modeling of 3D environments and UAV path planning problems, followed by a comparative analysis of IAPO’s performance against multiple algorithms in diverse 3D scenarios. Finally, Section 5 concludes the paper with a summary of key findings and future research directions.

## 2. APO Algorithm Overview

This section introduces the APO algorithm [30], explaining its mathematical model and providing a comprehensive analysis of its mechanisms. The symbols and nomenclature used in this section are presented in Table A1.

### 2.1. Traditional Artificial Protozoa Optimizer

The APO algorithm mimics the behaviors of euglena, specifically foraging, dormancy, and reproduction. The cell structure of the euglena is shown in Figure 1. These mechanisms balance exploration and exploitation during optimization. Autotrophic foraging and dormancy focus on exploration, while heterotrophic foraging and reproduction prioritize exploitation.

#### 2.1.1. Autotrophic Foraging

Euglena synthesize nutrients through chloroplasts, a process referred to as autotrophic foraging. Under strong light, the organism moves toward lower light intensity, exhibiting photophobia, while in low light, it moves toward higher intensity, exhibiting phototropism. The mathematical model for autotrophic foraging is as follows:(1)$$X_{i}^{new}=X_{i}+f\cdot \left(X_{j}-X_{i}+\frac{1}{np}\cdot \sum_{k=1}^{np}w_{a}\cdot \left(X_{k-}-X_{k+}\right)\right)⊙M_{f}^{\prime }$$(2)$$f=rand\cdot \left(1+\cos\left(\frac{iter}{iter_{max}}\cdot \pi \right)\right)$$(3)$$w_{a}=e^{-\left|\frac{f\left(X_{k-}\right)}{f\left(X_{k+}\right)+eps}\right|}$$(4)$$M_{f}\left[di\right]=\left\{\begin{array}{c} 1,\;\;\;\;if\;di\;is\;in\;rand\;perm(dim,\lceil dim\cdot \frac{i}{ps}\rceil )\\ 0,\;\;\;\;otherwise \end{array}\right.$$

Equation (1) demonstrates that in autotrophic foraging, position updates are governed by pairwise neighbor interactions, regulated by the autotrophic foraging coefficient $w_{a}$. The ranking index of the neighbor pair j<i implies that the ith euglena is at a lower light intensity and exhibits phototropism; if the neighbor pair has the ranking index j>i, then the ith euglena is at a higher light intensity and exhibits photophobia.

#### 2.1.2. Heterotrophic Foraging

In dark environments, euglena acquire nutrients by absorbing organic matter, a process called heterotrophic foraging. Assuming that a nutrient-rich location exists near $X_{i}^{new}$, the organism moves toward it. The position update is calculated as follows:(5)$$X_{i}^{new}=X_{i}+f\cdot \left(X_{near}-X_{i}+\frac{1}{np}\cdot \sum_{k=1}^{np}w_{h}\cdot \left(X_{i-k}-X_{i+k}\right)\right)⊙M_{f}$$(6)$$X_{near}=\left(1\pm Rand\cdot \left(1-\frac{iter}{iter_{max}}\right)\right)⊙X_{i}$$(7)$$w_{h}=e^{-\left|\frac{f\left(X_{i-k}\right)}{f\left(X_{i+k}\right)+eps}\right|}$$(8)$$Rand=\left[rand_{1},rand_{2},\ldots ,rand_{dim}\right]$$

#### 2.1.3. Dormancy

When exposed to environmental stress, euglena enter dormancy as a survival strategy. This process involves replacing dormant euglena with newly generated individuals to maintain population diversity and avoid local optima. The mathematical model of dormancy is as follows:(9)$$X_{i}^{new}=X_{min}+Rand⊙\left(X_{max}-X_{min}\right)$$(10)$$X_{min}=\left[lb_{1},lb_{2},\ldots ,lb_{dim}\right]\;X_{max}=\left[ub_{1},ub_{2},\ldots ,ub_{dim}\right]$$

It is the random emergence of domains that would be defined by the generation of new euglena through dormancy that qualifies the APO algorithm to jump out of local optima.

#### 2.1.4. Reproduction

When euglena reach optimal age and health conditions, they undergo asexual reproduction. While this process theoretically produces two identical offspring, our simulation implements this mechanism by generating a duplicate euglena with controlled perturbation. The mathematical model for this reproductive process is expressed as follows:(11)$$X_{i}^{new}=X_{i}\pm rand\cdot \left(X_{min}+Rand⊙\left(X_{max}-X_{min}\right)\right)⊙M_{r}$$(12)$$M_{r}\left[di\right]=\left\{\begin{array}{c} 1,\;\;\;\;if\;di\;is\;in\;rand\;perm(dim,\lceil dim\cdot rand\rceil )\\ 0,\;\;\;\;otherwise \end{array}\right.$$

#### 2.1.5. Algorithm Analysis

The advantage of the APO algorithm is its ability to dynamically adjust the ratio of exploitation to exploration so that the algorithm is biased towards exploration at the beginning and gradually shifts to exploitation as the number of iterations increases. The scheme is implemented as follows: Firstly, Equation (13) demonstrates that the initial state of the euglena is based on the scale fraction pf, which classifies the euglena as being involved in foraging (autotrophic or heterotrophic), dormant, or breeding. Autotrophic foraging and dormancy focus on large-scale searches, enhancing the algorithm’s exploration capabilities, while heterotrophic foraging and reproduction emphasize identifying promising regions, thereby improving exploitation efficiency. Secondly, the probability of autotrophic foraging versus heterotrophic foraging is regulated by a proportionality coefficient, $p_{ah}$. According to Equation (14), as the number of iterations increases, $p_{ah}$ decreases; thus, the euglena gradually shifts from autotrophic foraging to heterotrophic foraging. Finally, the probability of dormancy and reproduction is related to the ranking of the euglena. According to Equation (15), inferior euglena tend to enter dormancy to enhance their exploration. On the contrary, high-quality euglena tend to reproduce to enhance their exploitation.(13)$$pf=pf_{max}\cdot rand$$(14)$$p_{ah}=\frac{1}{2}\cdot \left(1+cos\left(\frac{iter}{iter_{max}}\cdot \pi \right)\right)$$(15)$$p_{dr}=\frac{1}{2}\cdot \left(1+cos\left(\left(1-\frac{i}{ps}\right)\cdot \pi \right)\right)$$

## 3. IAPO Algorithm

In this section, we introduce the specific improvements made to the IAPO algorithm. We tested the improved IAPO on the CEC2022 benchmark functions and conducted comparative experiments with nine other algorithms, followed by engineering applications and ablation experiments. The experimental results demonstrate that the improvements significantly enhance the algorithm’s global search ability, convergence speed, robustness, and adaptability.

### 3.1. Algorithm Improvement Scheme

#### 3.1.1. Population Initialization Based on Tent Map and Refractive Opposition-Based Learning

The tent map [31,32] is a well-known nonlinear chaotic mapping that generates pseudo-random sequences. It has broad applications in optimization, cryptography, and other scientific fields. It enhances population diversity and global search efficiency by modeling chaotic behavior with a simple piecewise linear function. The tent map is mathematically defined as follows:(16)$$x_{n+1}=\left\{\begin{array}{c} \frac{x_{n}}{a},\;\;\;\;\;\;\;\;if\;x_{n}<a\\ \frac{1-x_{n}}{1-a},\;\;\;\;if\;x_{n}\geq a\;\;\;\;\;\;\;\; \end{array}\right.$$
where $x_{n}\in \left[0,1\right]$ represents the current iteration value, and $a\in \left(0,1\right)$ is a control parameter, typically set to a=0.5 to achieve ideal chaotic properties. The tent map exhibits sensitivity to initial values and excellent uniform distribution characteristics, making it widely used in optimization problems for population initialization to improve the algorithm’s global exploration capability. Unlike other chaotic mappings, the tent map is easy to implement and computationally efficient, making it ideal for high-dimensional optimization.

ROBL [24,33,34,35] enhances the opposition-based learning (OBL) [36] strategy. While traditional OBL generates ‘oppositional solutions’ to expand search space exploration, which helps the optimization algorithm to explore a wider solution space more efficiently, ROBL introduces an additional refractive mechanism to further extend solution space coverage and improve algorithmic performance.

By introducing ROBL, the diversity of the population generated by the tent map is further enhanced. This ensures that the initial population of IAPO can better cover the solution space and provides a more favorable starting point for subsequent optimization processes.

The solutions generated by the ROBL method through the refraction operation are usually highly exploratory, allowing them to jump out of local optimal solutions and thus improve the global search capability of the optimization algorithm. In practical applications, particularly for high-dimensional complex optimization problems, ROBL demonstrates superior performance by improving the solution quality. This integration into IAPO’s initial population ensures better solution space coverage and establishes an optimal starting point for subsequent optimization.

In Figure 2, the x-axis divides the space into two parts, top and bottom, which are treated as different media. Suppose the search interval of the particle on the x-axis is $\left[a,b\right]$, that is, $x\in \left[a,b\right]$, and Y is the normal. A light source $x^{\prime }$ is located at a certain height directly above the particle as the point of incidence, and it emits an incident light ray $x^{\prime }O$ to the junction point O, with the length of the incident light ray being h. The light ray $x^{\prime }O$ undergoes refraction at the junction point O, and the refracted light ray is $Ox^{\prime *}$, with the length of the refracted light ray being $h^{*}$. Geometric relations can be used to find the incident angle θ and the refraction angle φ:(17)$$sin\theta =\frac{\left(\left(a+b\right)/2-x\right)}{h}$$(18)$$sin\varphi =\frac{\left(x^{*}-\left(a+b\right)/2\right)}{h^{*}}$$

The refractive index of the light can be found through Equations (17) and (18), as shown in Equation (19):(19)$$n=\frac{sin\theta }{sin\varphi }=\frac{\left(\left(\left(a+b\right)/2-x\right)/h\right)}{\left(\left(x^{*}-\left(a+b\right)/2\right)/h^{*}\right)}$$

The order $k=h/h^{*}$ can be obtained:(20)$$k\cdot n=\frac{\left(\left(a+b\right)/2-x\right)}{\left(x^{*}-\left(a+b\right)/2\right)}$$

From Equation (20), we obtain the following:(21)$$x^{*}=\frac{\left(a+b\right)/2-x}{k\cdot n}+\frac{a+b}{2}$$

When both n and k are equal to 1, Equation (21) can be reduced to the standard opposite learning formula:(22)$$x^{*}=a+b-x$$

It is evident that the OBL strategy represents a special case of the ROBL strategy. To enhance the diversity of the initialized population in the IAPO algorithm, the ROBL strategy is employed and extended to D-dimensional space, expressed as follows:(23)$$x_{i,j}^{*}=\frac{\left(a_{j}+b_{j}\right)/2-x_{i,j}}{k\cdot n}+\frac{a_{j}+b_{j}}{2}$$
where $X_{i,j}$ denotes the jth dimensional value of the ith individual, and $x_{i,j}^{*}$ is the position obtained by the ROBL strategy; typically, n=1 and k∈[0.5,1.5].

In the algorithm initialization process, we assume that the initial population is P=${\{X}_{1}$, …, $X_{N}\}$, where the individuals are randomly generated within the value range using random numbers. First, we apply a tent map to the population to obtain $P_{1}=\{X_{1}^{\prime },\ldots ,X_{N}^{\prime }\}$. Then, the population $P_{1}$ is updated using the ROBL strategy, resulting in $P_{1}^{*}$=$\left\{X_{1}^{*},\ldots ,X_{N}^{*}\right\}$. Next, we merge the two populations to form $P^{*}=\{X_{1}^{\prime },\ldots ,X_{N}^{\prime },X_{1}^{*},\ldots ,X_{N}^{*}\}$. Finally, we compute the fitness of each individual in the merged population $P^{*}$, and we select the top N individuals based on their fitness as the initial population for IAPO. This process ensures the diversity of the initial population, accelerates the convergence speed, and enhances the global search capability.

As illustrated in Figure 3, we demonstrate population distributions within a 600 m × 600 m two-dimensional space under a population size of 300, including a random population distribution, a tent map-processed distribution, and an ROBL-processed distribution. Notably, compared with the uneven random population distribution, the proposed tent map and ROBL methods achieve a more widespread and uniform spatial coverage of the population. This comparative analysis confirms the effectiveness of our method in enhancing population initialization.

#### 3.1.2. Dynamic Optimal Leadership Mechanism

In the standard APO algorithm, individuals update their positions during autotrophic foraging by comparing fitness values at random points and considering neighbor influence. While this enables a global search, it has limitations in terms of its search efficiency and convergence speed. To address these issues, a dynamic optimal leadership mechanism is introduced to accelerate global convergence by guiding the population toward the optimal solution. In each iteration, the optimal leadership is identified as the individual with the highest fitness, and its position represents the optimal solution in the current search space. The improvement strategy can be expressed mathematically as follows:(24)$$X_{i}^{new}=X_{i}+\left(L_{i}-X_{i}\right)+f\cdot \left(X_{j}-X_{i}+\frac{1}{np}\cdot \sum_{k=1}^{np}w_{a}\cdot \left(X_{k-}-X_{k+}\right)\right)⊙M_{f}$$
where $L_{i}$ is the position of the most adaptive euglena in the population obtained after each iteration. In the improved IAPO, this mechanism reshapes the euglena’s search behavior during autotrophic foraging. Each euglena not only updates its position based on its own and neighboring individuals’ positions but is also influenced and guided by the leader’s position. This approach directs the population toward more promising search areas, thereby accelerating the algorithm’s convergence.

To prevent premature convergence due to the leader’s influence, we introduce a nonlinear dynamic adjustment factor to ensure sufficient exploration. This factor dynamically modulates the influence of the optimal leader on individual updates. The nonlinear dynamic adjustment factor is designed to exhibit small and slowly varying values in the early stages of the algorithm’s iteration, while demonstrating larger and rapidly changing values in the later stages. This design effectively balances global exploration and local exploitation, preventing the algorithm from prematurely falling into local optima while accelerating convergence in the later phases. The nonlinear dynamic adjustment factor is formulated as follows:(25)$${LF}_{i}=\frac{e^{\alpha \cdot \frac{i}{{iter}_{max}}}\;-1}{e^{\alpha }-1},i=1,2,\ldots ,{iter}_{max}$$
where LF represents the nonlinear dynamic adjustment factor, ${iter}_{max}$ is the maximum number of iterations, and α is the adaptive convergence factor. By tuning α, the evolution curve of the leader’s influence can be precisely controlled, and α should be adjusted according to the maximum number of iterations. For instance, when ${iter}_{max}=500\;and\;\alpha =3$, the curve of ${LF}_{i}$ is as shown in Figure 4a.

With the introduction of LF, the autotrophic foraging in Equation (23) is modified to account for the dynamic influence of the leader:(26)$$X_{i}^{new}=X_{i}+{LF}_{i}\cdot \left(L_{i}-X_{i}\right)+f\cdot \left(X_{j}-X_{i}+\frac{1}{np}\cdot \sum_{k=1}^{np}w_{a}\cdot \left(X_{k-}-X_{k+}\right)\right)⊙M_{f}$$

This dynamic optimal leadership mechanism is introduced into the IAPO algorithm’s self-feeding process through the leader and the nonlinear dynamic adjustment factor. From the variation curve of the nonlinear dynamic adjustment factor, we can observe that this factor is relatively small in the early iterations, ensuring that individuals in the population primarily rely on self-exploration, maintaining a broad search of the solution space and avoiding local optima. In the later iterations, the factor gradually increases, enhancing the influence of the optimal value on exploration. The guidance of the optimal value attracts more primitive organisms toward the optimal solution, thereby accelerating population convergence. The integration of these two mechanisms improves both the convergence speed and the solution accuracy of the algorithm.

#### 3.1.3. Cauchy Mutation Strategy

The Cauchy distribution [37,38] is characterized by its heavy-tailed properties, with its probability density function decaying more slowly at the tails compared to the normal distribution. This implies that the Cauchy distribution is more likely to generate random numbers with large offsets, thereby enabling individuals in the population to leap to distant regions in the search space. This feature is particularly beneficial for solving high-dimensional optimization problems and avoiding premature convergence of the population to local optima. Specifically, during the update of individual positions, a random factor generated from the Cauchy distribution (referred to as the Cauchy mutation factor) can be introduced to maintain larger variations during the exploitation process. The probability density function of the standard Cauchy distribution is illustrated in Figure 4b and is mathematically expressed as follows:(27)$$f\left(x\right)=\frac{1}{\pi \left(1+x^{2}\right)},x\in \left(-10,10\right)$$

In the reproduction phase, we design the Cauchy mutation factor as ‘Cy’ to perturb and mutate the current solutions, thereby generating new solutions and reducing the risk of the original algorithm falling into local optima:(28)$$\;\mathrm{Cy}\;=1/2+1/2\tan\left(0.5\times \pi \times \left(\;\mathrm{rand}\;-0.5\right)\right)$$

After the incorporation of the Cauchy mutation factor, the update formula for the reproduction phase can be expressed as follows:(29)$$X_{i}^{new}=X_{i}\pm Cy\cdot \left(X_{min}+Rand⊙\left(X_{max}-X_{min}\right)\right)⊙M_{r}$$

From the latest position update formula, we deduce that incorporating a Cauchy factor in the reproduction phase dynamically adjusts the position updates. In the early iterations, this ensures diversity in the population updates, while in the later iterations, the heavy-tailed characteristic of the Cauchy distribution maintains a certain level of update capability while restricting the distance of new positions. This mechanism guarantees that the algorithm can escape local optima during the exploration process, thereby increasing the probability of finding the global optimum.

#### 3.1.4. Simulated Annealing Algorithm

The simulated annealing (SA) algorithm [39] is a widely used global optimization method inspired by the annealing process of solids in physics. Annealing refers to the process of obtaining a crystal structure with the lowest energy by gradually lowering the temperature of a substance until thermal equilibrium is reached. The SA algorithm utilizes this principle for optimization, making it particularly effective for solving complex problems with multiple local optima. In SA, each iteration’s solution is accepted or rejected based on a probability function. The core mechanism of SA is the ‘temperature parameter’ T, which decreases as the number of iterations increases. When the temperature is high, the algorithm allows the acceptance of poorer solutions, enhancing its global search capability. As the temperature decreases, the algorithm focuses on high-quality solutions, eventually converging to the global optimum. The temperature parameter T is designed as follows:(30)$$T\left(i\right)=T_{\mathrm{initial}}\cdot e^{-\beta \cdot i},i=1,2,\ldots ,{iter}_{max}$$
where $T_{\mathrm{initial}}$ represents the initial temperature, set as $T_{\mathrm{initial}}=100$. The parameter β is the temperature decay coefficient, which is determined based on ${iter}_{max}$. For example, when ${iter}_{max}=500$ and β=0.01, the temperature change curve is as shown in Figure 4c. With the variation of the temperature parameter T, the acceptance probability of solution updates is calculated as follows:(31)$$P=\left\{\begin{array}{c} 1,\;\;\;\;\;\;\;\;\;\Delta E<0\\ e^{-\frac{\Delta E}{T}}\;,\;\;\;\Delta E>0 \end{array}\right.$$
where ΔE represents the difference between the fitness of the updated solution and that of the original solution. If the updated solution is better, it is accepted directly; if worse, it is accepted with a probability proportional to ΔE and T, allowing the algorithm to escape local optima.

In the heterotrophic foraging and dormancy phases of the IAPO algorithm, we integrate the SA algorithm. The difference between the fitness of the newly updated position and that of the original position, denoted by ΔE, is used as a parameter to compute the acceptance probability. Under the influence of the SA algorithm, inferior solutions are accepted with a certain probability in each iteration, and this probability gradually decreases as the temperature decay coefficient decreases. By incorporating SA, the IAPO algorithm not only enhances its ability to escape local optima but also improves its convergence performance. Additionally, the solution exploration strategy becomes more flexible, allowing the algorithm to fully utilize the guidance of the current optimal solution while avoiding blind updates of population individuals, ultimately improving the accuracy and stability of the final solution.

#### 3.1.5. Computational Complexity

The proposed IAPO algorithm primarily performs sorting, autotrophic and heterotrophic foraging, dormancy and reproduction, and fitness function evaluations. The time complexity of the initialization process consists of two parts: fitness value computations and population sorting. The time complexity of the initialization process can be expressed as $2O\left(ps\cdot f\left(\cdot \right)\right)+O\left(2ps\cdot log\left(2ps\right)\right)$, that of autotrophic and heterotrophic foraging is $O\left(\left(ps-\lceil ps\cdot pf\rceil \right)\cdot np\cdot dim\right)$, that of dormancy and reproduction is $O\left(\lceil ps\cdot pf\rceil \cdot dim\right)$, and that of fitness function evaluations is $O\left(ps\cdot f\left(\cdot \right)\right)$, Consequently, the total complexity is $O\left(2O\left(ps\cdot f\left(\cdot \right)\right)+O\left(2ps\cdot log\left(2ps\right)\right)+{iter}_{max}\cdot \left(ps\cdot log\left(ps\right)+\left(\left(ps-\lceil ps\cdot pf\rceil \right)\cdot np+\lceil ps\cdot pf\rceil \right)\cdot dim+ps\cdot f\left(\cdot \right)\right)\right)$.

#### 3.1.6. Algorithm Pseudocode

Pseudocode of the multi-strategy fusion of artificial protozoan optimizer see in Algorithm 1.
**Algorithm 1****Input:** Initialize parameters ps, dim, np, itermax, pfmax, k, α, β, Tinitial, and MaxFEs (maximum function evaluations).**Output:** The global optima Xgbest and f(Xgbest).1 Initialize population P = {X1,…, XN}2 **for** i = 1 → ps **do**3  xn+1=xna,        if xn<a1−xn1−a,    if xn≥a, P1=X1′,…,XN′  // tent map4  xi,j*=aj+bj/(2−xi,j)/k+(aj+bj)/2, j = 1, 2,…,dim // ROBL5  P1*=X1*,…,Xps*, P*=X1′,…,XN′,X1*,…,XN*   // initial population consolidation6  sort(Pi*), i = 1, 2,…, 2ps, X ={X1, …, Xps}  // screened populations according to fitness7 **end for**8 **while** FEs < MaxFEs **do**  // check whether the maximum number of iterations is reached9    sort(Xi), i = 1, 2,…, ps; 10  pf=pfmax;  // proportion fraction11  Drindex=randperm(ps,⌈ps⋅rand⌉)12  **for** i = 1: ps **do**  // ergodic population13    **if** i is in Drindex **then**14      **if** Pdr>rand **then**15        Calculate Xi* using Equation (10)  // dormancy16        Calculate ΔE=f(Xi*)−f(Xi)  // calculate the difference in fitness17        Calculate P using Equation (27)  // probability of acceptance18         **if** rand<P **then**
Xinew=Xi*19        **else** Xinew=Xi20        **end if**21      **else**22        Mf[1,randperm(dim,⌈dim⋅rand⌉)]=123        Calculate Xi* using Equation (11)  // reproduction24        Calculate ΔE=f(Xi*)−f(Xi)  // calculate the difference in fitness25        Calculate P using Equation (27)  // probability of acceptance26        **if** rand<P **then**
Xinew=Xi*27        **else** =Xi28        **end if**29      **end if**30    **else**31      Mf[1,randperm(dim,⌈dim,i/ps⌉)]=132      **if** Pah>rand **then**33        Calculate LF using Equation (24)  // nonlinear dynamic adjustment factor34        Calculate Xi* using Equation (25)  // foraging by an autotroph35      **else**
36        Calculate Cy using Equation (24)  // calculate the Cauchy factor37        Calculate Xinew using Equation (5)  // foraging by a heterotroph38      **end if**39    **end if**40    **if** f(Xinew)<f(Xi) **then** Xi←Xinew41    **else** Xi←Xi42    **end if**43  **end for**44  Xgbest=opt{Xi}  // update the optimal solution45  FEs←FEs+ps46  **end while**

### 3.2. Experiments and Analyses

This section presents the experimental validation of the improved IAPO algorithm. In Section 3.2.1, we introduce the experimental environment configuration. In Section 3.2.2, we verify the effectiveness of the improvements by comparing the performance of the proposed algorithm with that of nine other algorithms on 12 benchmark functions from CEC2022. In Section 3.2.3, we analyze the variation trends of exploration and exploitation by performing equation-based calculations on the 12 benchmark functions. In Section 3.2.4, we conduct engineering applications and ablation experiments to further evaluate the algorithm.

#### 3.2.1. Development Environment Settings

All experiments were conducted on a laptop equipped with the Windows 11 operating system, 32 GB of RAM, and an AMD Ryzen Al 9 365 @ 2.0 GHz processor. MATLAB 2024a was used as the development platform for implementing the algorithm and conducting tests.

#### 3.2.2. CEC2022 Benchmark Function Test Results and Analysis

The performance of the proposed IAPO algorithm was evaluated using 12 benchmark functions from CEC2022 [40]. These functions include unimodal functions, multimodal functions, hybrid functions, and composition functions. The detailed definitions of these benchmark functions are provided in Table 1. IAPO’s stability and convergence speed were compared against those of nine other algorithms, demonstrating its superior performance and confirming the effectiveness of the proposed improvements.

In the improved IAPO algorithm, NP was set to 1, and ${pf}_{max}$ was set to 0.1, considering the algorithm’s complexity. To validate the superiority of the improved IAPO, comparisons were conducted against several optimization algorithms, including the artificial protozoa optimizer (APO) [30], the crested porcupine optimizer (CPO) [41], the dung beetle optimizer (DBO) [42], particle swarm optimization (PSO) [43], grey wolf optimization (GWO) [44], ant colony optimization (ACO) [45], Harris hawks optimization (HHO) [46], the optical microscope algorithm (OMA) [47], and the sparrow search algorithm (SSA) [48].

To ensure experimental fairness, identical parameter settings were applied across all algorithms. Specifically, the function dimension was set to dim=20, the population size was set to ps=50, the maximum number of iterations was ${iter}_{max}=500$, and each algorithm was executed 30 times to enhance the experimental reliability.

The algorithms’ performance was assessed using the mean and standard deviation, ranking them based on their average performance over 12 CEC2022 benchmark functions. These metrics quantify optimization capability, convergence, and stability. Table 2 shows that IAPO outperformed APO on eleven benchmark functions and performed similarly on the remaining one. This confirms the effectiveness of the proposed improvements. Compared to the other algorithms, IAPO ranked first on eight benchmark functions (F1, F4–F8, F10, and F11), second on three functions (F3, F9, and F12), and third on F2. These results underscore IAPO’s strong optimization performance, superior stability, and accuracy.

Figure 5 visually illustrates IAPO’s superior performance through convergence curves on CEC2022 test functions. The IAPO algorithm curve is shown in red, while the APO algorithm curve is in green. The curves show that IAPO’s fitness values dropped quickly and reached lower levels at the maximum iteration, outperforming most algorithms.

The data analysis and visualization result further validate IAPO’s excellent performance in multiple test scenarios. By integrating the advantages of multiple strategies, the IAPO algorithm exhibits outstanding global search capabilities and local exploitation efficiency, achieving superior optimal solutions and convergence speeds across a variety of test functions. These results highlight IAPO’s potential as an innovative and effective optimization algorithm.

#### 3.2.3. Exploration and Development Analyses

The convergence of metaheuristic algorithms is fundamentally determined by the balance between exploration and exploitation. Exploration broadly searches the solution space for better solutions and avoids local optima. Exploitation intensively searches promising regions to accelerate convergence and refine the optimal solution.

Striking a balance between exploration and exploitation is critical in algorithm design. An excessive emphasis on exploration may lead to prolonged searches in unexplored regions, hindering the algorithm from effectively leveraging existing information to converge to the optimal solution. This imbalance often results in slow convergence or failure to find the desired solution. Conversely, excessive exploitation risks premature convergence, limiting exploration and leading to suboptimal results.

To comprehensively evaluate the performance of the improved IAPO algorithm, we introduced the dimensional diversity metric. By calculating the exploration and exploitation rates [30], we analyzed the algorithm’s ability to balance these two aspects effectively. The corresponding formulas are as follows:(32)$$err=\frac{div}{div_{max}}$$(33)$$eir=\frac{div_{max}-div}{div_{max}}$$(34)$$div=\frac{1}{dim\cdot ps}\cdot \sum_{j=1}^{dim}\sum_{i=1}^{ps}\left|median\left(x^{j}\right)-x_{i}^{j}\right|$$
where err and eir represent the exploration and exploitation rates in an iteration, respectively. The population diversity of a single generation is denoted by div, while ${div}_{max}$ represents the maximum diversity across all generations. The median value of all individuals in the jth dimension is represented by $median\left(x^{j}\right)$, and $x_{i}^{j}$ denotes the value of the ith individual in the jth dimension.

To analyze the dynamics of exploration and exploitation rates, the experiment set the population size to 50 and evaluated the 12 benchmark functions of CEC2022 over 500 iterations. The experimental results of the IAPO algorithm are shown in Figure 6. During the initial iteration phase, the IAPO algorithm emphasizes exploration while maintaining a relatively low exploitation level. As the iterations progress, the algorithm gradually reduces its extensive exploration of the solution space, shifting focus to exploiting potentially superior solution regions.

This is attributed to our dynamic optimal leadership strategy. In the early stage of the algorithm, a relatively small nonlinear dynamic adjustment factor reduces the influence of the current best solution on the exploration capability, thereby allowing the algorithm to thoroughly search the entire solution space. As time progresses, the nonlinear dynamic adjustment factor gradually increases, which, in turn, enhances the influence of the optimal solution on each euglena. This shift facilitates a transition from exploration to exploitation, endowing the algorithm with robust exploitation capabilities during the later stages of iteration. Overall, the IAPO algorithm achieves a favorable dynamic balance between exploration and exploitation throughout the iteration process, further demonstrating its effectiveness and robustness in optimization tasks.

#### 3.2.4. Engineering Applications and Ablation Experiments

Currently, swarm intelligence algorithms are widely used to solve various engineering optimization tasks. In this study, we selected two engineering optimization problems for testing and conducted ablation experiments by selectively removing specific improvement strategies or mechanisms. The purpose of this research was to evaluate the impact of the retained strategies on the algorithm and to determine the contribution and importance of each component within the IAPO algorithm framework. In these experiments, the population size was set to ps=50 and the maximum number of iterations was set to ${iter}_{max}=500$.

We designed the following variants:

IAPO_I: Based on IAPO, the tent map and ROBL population initialization strategies were removed.

IAPO_II: Based on IAPO, the dynamic optimal leadership strategy was removed.

IAPO_III: Based on IAPO, the Cauchy mutation strategy was removed.

Welding beam design

The welding beam design is an optimization problem aimed at minimizing manufacturing costs. The core of the problem is to determine the four design variables—beam length l, height t, thickness b, and weld thickness h—that satisfy all constraints, including the given shear stress τ, bending stress θ, critical bending load $P_{c}$ for the beam, end displacement δ, and boundary conditions, so that the total manufacturing cost of the welded beam is minimized. Since this problem involves multiple nonlinear constraints and an objective function simultaneously, the WBD problem can be regarded as a typical nonlinear programming problem. These parameters are represented as the vector $\vec{x}=\left[x_{1,}x_{2,}x_{3}{,x}_{4}\right]=\left[h,l,t,b\right]$, with its mathematical formulation as follows:(35)$$f\left(\vec{x}\right)=1.10471{\cdot x}_{1}^{2}{\cdot x}_{2}+0.04811{\cdot x}_{3}{\cdot x}_{4}\cdot \left(14.0+x_{2}\right)$$

The constraints are as follows:(36)$$\begin{array}{c} g_{1}\left(\vec{x}\right)=\tau \left(\vec{x}\right)-\tau_{max}⩽0,\\ g_{2}\left(\vec{x}\right)=\sigma \left(\vec{x}\right)-\sigma_{max}⩽0,\\ g_{3}\left(\vec{x}\right)=\delta \left(\vec{x}\right)-\delta_{max}⩽0,\\ g_{4}\left(\vec{x}\right)=x_{1}-x_{4}⩽0,\\ g_{5}\left(\vec{x}\right)=P-P_{c}\left(\vec{x}\right)⩽0,\\ g_{6}\left(\vec{x}\right)=0.125-x_{1}⩽0,\\ g_{7}\left(\vec{x}\right)=1.10471{\cdot x}_{1}^{2}{\cdot x}_{2}+0.04811\cdot x_{3}{\cdot x}_{4}\cdot \left(14.0+x_{2}\right)-5.0⩽0 \end{array}$$

The boundary conditions are as follows:(37)$$\begin{array}{c} 0.1⩽x_{1}⩽2,\\ 0.1⩽x_{2}⩽10,\\ 0.1⩽x_{3}⩽10,\\ 0.1⩽x_{4}⩽2 \end{array}$$

Table 3 presents the welding beam design optimization results under different conditions.

2.Tension/compression spring design

The objective of the tension/compression spring design is to minimize the spring mass $f\left(x\right)$ while satisfying specific constraints. The constraints include four inequalities: minimum deflection, shear stress, oscillation frequency, and outer diameter limitations. The design variables consist of three components: the mean coil diameter $D\left(x_{2}\right)$, the spring wire diameter $d\left(x_{1}\right)$, and the effective number of coils $N\left(x_{3}\right)$. The detailed mathematical model is given by the following equation:(38)$$f\left(x\right)=\left(N+2\right)\cdot D{\cdot d}^{2}$$

The constraints are as follows:(39)$$\begin{array}{c} g_{1}\left(x\right)=1-\frac{D^{3}\cdot N}{71785\cdot d^{4}}\leq 0,\\ g_{2}\left(x\right)=\frac{4{\cdot D}^{2}-d\cdot D}{12566\left({D\cdot d}^{3}-d^{4}\right)}+\frac{1}{5108\cdot d^{2}}-1\leq 0,\\ g_{3}\left(x\right)=1-\frac{140.45\cdot d}{D^{2}\cdot N}\leq 0,\\ g_{4}\left(x\right)=\frac{D+d}{1.5}-1\leq 0 \end{array}$$

The boundary conditions are as follows:(40)$$\begin{array}{c} 0.05⩽x_{1}⩽2,\\ 0.25⩽x_{2}⩽1.3,\\ 2⩽x_{3}⩽15 \end{array}$$

Table 4 presents the optimization results for the tension/compression spring design under different conditions.

After conducting ablation experiments on these two engineering optimization problems, we found that the population initialization method using the tent map and ROBL significantly improved the quality of the initial population. In optimization problems, a higher-quality initial population can help the algorithm search for the optimal solution earlier and more accurately. The dynamic optimal leadership strategy and Cauchy mutation strategy we designed complement each other: the former guides the population’s search process to continuously approach the current optimal solution, greatly accelerating convergence and enhancing exploration of the optimum, while the latter boosts the algorithm’s ability to escape local optima, ensuring robust exploration. As for the integrated SA algorithm, although it does not enhance the exploration capability, its probabilistic acceptance of inferior solutions reduces their adverse impact on convergence speed while still allowing the algorithm to escape local optima.

Through these comparative experiments, we conclude that our proposed approach enhances both the development and exploration capabilities of the algorithm. The dynamic adjustment scheme better balances these two aspects, and under various experimental conditions, the IAPO algorithm consistently outperforms the original algorithm, demonstrating strong optimization ability and robustness.

## 4. Overview of UAV Path Planning

This section presents the modeling of peaks in the UAV path planning problem, including the physical model of the path planning problem and the design of four different peak environments. To evaluate algorithm performance, we use a simple map with 6 peaks and a complex map with 14 peaks to compare the improved IAPO algorithm with nine other algorithms, demonstrating its superiority in solving the path planning problem.

### 4.1. Problem Modeling

#### Three-Dimensional Environment Modeling

In real-world scenarios, UAV path planning occurs in three-dimensional space. According to the literature [20], the three-dimensional space can be modeled using Equation (41). Considering that the real geographical environment often includes restricted areas such as crowded areas and military no-fly zones, it is necessary to establish a corresponding no-fly zone model [49]. Here, the no-fly zone is modeled as a cylinder.(41)$$Z\left(x,y\right)=\sum_{i=1}^{N}h_{i}exp\left[-{\left(\frac{x-x_{i0}}{x_{si}}\right)}^{2}-{\left(\frac{y-y_{i0}}{y_{si}}\right)}^{2}\right]$$(42)$$B=\left[\begin{array}{cccc} x_{1} & y_{1} & r_{1} & h_{1}\\ x_{2} & y_{2} & r_{2} & h_{2}\\ \ldots & & & \\ x_{n} & y_{n} & r_{n} & h_{n} \end{array}\right]$$

Using Equation (41), the total number of threat points, N; the height of each threat point, $h_{i}$; the coordinates of the center of each threat point, $\left(x_{i0},y_{i0}\right)$; and the slopes of the threat points along the X and Y axes, $x_{si}$ and $y_{si}$, are defined. Using Equation (42), the no-fly zone’s center coordinates ${(x}_{i},y_{i})$, radius $r_{i}$, and height $h_{i}$ are specified. By combining these two equations, we construct a 600 m × 600 m × 100 m map with 14 peaks and a no-fly zone. The path starts at point ‘o’ and ends at point ‘*’ as shown in Figure 7.

### 4.2. UAV Path Planning Modeling

In complex UAV path planning scenarios, the primary objective is to design a safe, efficient, and feasible flight path that enables the UAV to navigate complex terrain and multiple obstacles with minimal energy consumption. To achieve this, the problem is formulated as a multi-constraint optimization task, incorporating constraints related to energy consumption minimization, safety assurance, and flight attitude feasibility. To address these constraints comprehensively, we construct a total cost function [20,49,50,51] that fully captures the optimization objectives and guides the path planning process:(43)$$f=w_{1}\cdot L+w_{2}\cdot H+w_{3}\cdot Curve$$
where f represents the total cost function to be optimized in the path trajectory problem, while L, H, and Curve denote the costs associated with path length, height, and curvature, respectively. The parameters $w_{1},\;w_{2},\;and\;w_{3}$ are the corresponding weights of these components.

If the numerical ranges of L, H, and Curve differ significantly (for example, if the path length is several hundred meters while the turning angle score is only in the single digits), then directly applying weighted summation may cause one metric to disproportionately influence the fitness, thereby affecting the optimization outcome. Normalization makes the fitness calculation more general and ensures that the optimization algorithm works effectively across different scenarios. We employ the Min–Max normalization method to scale all data to the [0,1] range:(44)$$X_{norm}=\frac{X-X_{min}}{X_{max}-X_{min}}$$

#### 4.2.1. Initial Conditions and Collision Detection

Before initiating path planning, it is essential to determine the initial condition, ensuring that all points are within the map’s range:(45)$$\left\{\begin{array}{c} x_{min}\leq x_{i}\leq x_{max}\\ y_{min}\leq y_{i}\leq y_{max}\\ z_{min}\leq z_{i}\leq z_{max} \end{array}\right.$$

In UAV path planning, path safety is one of the key issues, especially in complex environments where the path may collide with obstacles. To ensure the feasibility of the planned path, we introduce a collision detection mechanism. First, the Euclidean distance between the 3D coordinates of the path points and the center of the threat area is calculated; if this distance is less than the radius of the threat area, a collision is considered to have occurred, and the trajectory generated in that iteration is marked as invalid. Second, the corresponding height of the path points is obtained from the terrain elevation map, and it is determined whether the actual height of the path points is below the terrain. If a height collision occurs, the path is similarly marked as invalid. Only when the planned points are valid do we proceed with subsequent cost calculations. This detection method verifies the safety of the path on a point-by-point basis, effectively preventing the path from intersecting with obstacles or the terrain.

The collision detection mechanism is defined as follows:(46)$$\left\{\begin{array}{c} d_{k}=\sqrt{(x_{i}-x_{k}{)}^{2}+(y_{i}-y_{k}{)}^{2}}\\ d_{Z}=Z_{i}-Z_{interp}\\ SuccessFlag=\left\{\begin{array}{cc} 1, & d_{k}>d_{r}ord_{Z}>0\\ 0, & otherwise \end{array}\right.\;\;\;\; \end{array}\right.$$
where $d_{k}$ represents the distance from the current point to the center of each obstacle or no-fly zone, $d_{r}$ denotes the radius of the obstacle or no-fly zone, and $d_{Z}$ indicates the height difference between the current point and the terrain or no-fly zone. If $d_{k}>d_{r}\;or\;d_{Z}>0$, the current point is deemed free of collisions and can be used as a track point for path planning. Otherwise, it cannot be used as a track point. The total cost function f is computed only when SuccessFlag=1.

#### 4.2.2. Path Length Costs

In UAV path planning, the flight path L is divided into n waypoints, where the position of each waypoint is represented in 3D coordinates as $P_{i}=\left(x_{i},y_{i},z_{i}\right)$. The Euclidean distance between neighboring waypoints defines the length of each segment, and the total path length is calculated by summing the distances between consecutive waypoints. This total length serves as the length constraint in path planning.(47)$$L=\sum_{i=1}^{n}\sqrt{{\left(x_{i}-x_{i-1}\right)}^{2}+{\left(y_{i}-y_{i-1}\right)}^{2}+{\left(z_{i}-z_{i-1}\right)}^{2}}$$

#### 4.2.3. Curvature Constraint Costs

During flight, excessive diving, climbing, or turning angles can cause the UAV to stall, leading to mission failure. Assuming that the UAV’s current position is $\left(x_{i},y_{i},z_{i}\right)$ and its position at the next moment is $\left(x_{i+1},y_{i+1},z_{i+1}\right)$, the flight path must satisfy the curvature constraint, defined as follows:(48)$$C_{i}=\frac{\left(dx_{i}\cdot dx_{i+1}\right)+\left(dy_{i}\cdot dy_{i+1}\right)+\left(dz_{i}\cdot dz_{i+1}\right)}{\sqrt{dx_{i}^{2}+dy_{i}^{2}+dz_{i}^{2}}\cdot \sqrt{dx_{i+1}^{2}+dy_{i+1}^{2}+dz_{i+1}^{2}}}$$(49)$$Curve=\sum_{i=1}^{n-2}\left(cos\left(\phi \right)-C_{i}\right)$$
where $C_{i}$ represents the cosine of the angle between the path vector at waypoint i and the vector at waypoint i+1. The variable n denotes the total number of waypoints, and n−2 represents the number of neighboring vector pairs considered in the curvature calculation. The target angle ϕ is set to π/2.

#### 4.2.4. Cost of Height Variation

Altitude limits are critical to the execution and safety of UAV missions. Rapid changes in altitude can significantly increase energy consumption and lead to mission failure, while slow altitude adjustments heighten the risk of collisions with terrain or obstacles. To balance these factors, we introduce an altitude constraint cost:(50)$$H=\sum_{i=1}^{n}\sqrt{{\left(z_{i}-\bar{z}\right)}^{2}}$$
where $z_{i}$ represents the altitude of each waypoint, and $\bar{z}$ denotes the average altitude of all waypoints. The altitude cost H ensures that variations in altitude remain within acceptable limits, enhancing the feasibility and safety of the path.

### 4.3. Comparison of Various Algorithms for UAV Path Planning

Based on the environmental modeling described above, we conducted comparative simulation tests of UAV path planning in 3D terrain environments of varying complexity to comprehensively evaluate the performance of the improved IAPO algorithm. The simulations verified the adaptability and robustness of the IAPO algorithm across multiple complex scenarios.

#### 4.3.1. Environmental Settings

To ensure experimental fairness, identical initial conditions were established for each algorithm. We compared the improved IAPO algorithm with nine other algorithms—APO, CPO, DBO, PSO, GWO, ACO, HHO, OMA, and SSA—on the same simulation maps. Four different maps were designed: Map I and Map II are simple maps with six peaks. The start and end points for Map I are (10,10,10) and (490,530,20), respectively. For Map II, the start and end points are (10,10,10) and (490,530,50). Map III and Map IV are complex maps with 14 peaks, sharing the same start and end points as Map II. All four maps include a no-fly zone, further increasing the complexity of the planning task.

#### 4.3.2. Algorithm Comparison Experiments

To ensure consistent and reliable results, identical parameter settings were applied across all experiments, and each scenario was repeated independently 30 times. The evaluation metrics included the standard deviation, mean, and ranking, derived from the total cost computed during algorithm iterations. The total cost encompassed the path length, curvature constraints, and height variation costs, providing a comprehensive measure of algorithm performance.

For all tests, the population size was set to ps = 30, the maximum number of iterations to ${iter}_{max}=100$, and the parameter α to 3. Table 5 summarizes the results of the 10 algorithms across the four maps, while Figure 8 displays boxplots with algorithm names on the horizontal axis and fitness values on the vertical axis. These results highlight the superior performance of the improved IAPO algorithm across diverse scenarios.

The performance of traditional algorithms such as APO, ACO, and SSA exhibited significant fluctuations across the different scenarios. These variations are primarily due to the challenges in exploring candidate solutions within the solution space, arising from the introduction of multiple constraints and complex obstacle distributions. Furthermore, these algorithms lack sufficient adaptability to handle complex constraints, resulting in rankings that vary with changes in scenario complexity.

The improved IAPO algorithm demonstrated enhanced adaptability and robustness. According to the experimental results, its average fitness and standard deviation ranked first across the four different maps we designed. This performance is attributed to its balanced mechanism between local search and global optimization, which enables it to quickly pinpoint the optimal solution. In three-dimensional environments, the IAPO algorithm efficiently handles various constraints and maintains stable performance regardless of scene complexity. Under these settings, we also recorded the optimal route and the fitness curve in the three-dimensional environment after optimization, further validating the algorithm’s effectiveness.

Figure 9, Figure 10, Figure 11 and Figure 12 illustrate the performance of each algorithm in the path planning task across various scenarios. Through these comparisons, it can be observed that although all the tested algorithms accomplished the path planning tasks, there were significant differences in key metrics such as their path quality, convergence speed, and stability. The test results show that the proposed IAPO algorithm outperformed the other benchmark algorithms on the different maps we designed, indicating that the IAPO algorithm possesses higher adaptability and robustness when dealing with path planning problems under multiple constraints.

In summary, the superior performance of IAPO in path planning is attributed to its innovative improvements. The tent map and ROBL initialization strategies provide higher-quality initial solutions for the path planning problem; the dynamic optimal leadership strategy guides the euglena toward the optimal solution, enabling the algorithm to find the best path more quickly; and the nonlinear dynamic adjustment factor dynamically modulates the influence of the optimal solution on other euglena, ensuring robust early-stage exploration. The Cauchy mutation strategy leverages the heavy-tailed property of the Cauchy distribution to prevent the algorithm from becoming trapped in local optima, and the incorporation of the simulated annealing algorithm reduces the negative impact of inferior solutions on the convergence speed; its dynamic nature accelerates convergence while maintaining the ability to escape local optima. These improvements achieve an effective balance between global search and local optimization, enabling the algorithm to find the optimal path in complex three-dimensional spaces both faster and more accurately.

## 5. Conclusions

In this study, we proposed a multi-strategy integrated IAPO algorithm to address the three-dimensional path planning problem for UAVs in complex environments. In the algorithm design, we introduced the tent map and ROBL mechanisms to enhance the population diversity during initialization. Additionally, we developed a dynamic optimal leadership strategy to accelerate convergence while maintaining exploration capabilities, thereby preventing premature convergence during the search process. Furthermore, we incorporated a Cauchy mutation factor into the reproduction phase to enable population individuals to leap within the search space, avoiding entrapment in local optima. Finally, we integrated the simulated annealing algorithm into the heterotrophic foraging and dormancy phases, improving the convergence speed while ensuring the ability to escape local optima.

To validate the proposed improvements, we conducted comparative experiments on the CEC2022 benchmark functions, demonstrating the significant advantages of the IAPO algorithm in solving optimization problems. The experimental results show that IAPO algorithm effectively retains high-quality individuals throughout the iterative process while maintaining a dynamic balance between exploration and exploitation. Ultimately, we applied the IAPO algorithm to three-dimensional path planning problems and compared its performance with that of other commonly used algorithms. The experiments confirmed that the proposed method is versatile and efficient in generating low-cost, safe, and smooth UAV paths.

Moreover, the application of this research method holds significant managerial and operational value. First, in the logistics sector, this algorithm can provide precise path planning for UAV delivery systems, thereby reducing transportation costs, improving delivery efficiency, and effectively handling complex road conditions. Second, in disaster management and emergency rescue, this method can be used to plan the optimal routes for search-and-rescue UAVs, ensuring rapid and accurate arrival at accident sites in emergencies, thus enhancing response speed and safety. Finally, in precision agriculture, the proposed path planning enables UAVs to efficiently cover farmland, optimize operational routes, minimize redundancy and omissions, and improve both operational efficiency and resource utilization.

Future work will further explore the application of this algorithm in dynamic obstacle avoidance and UAV swarm path planning to address more complex and variable real-world scenarios.

## Figures and Tables

**Figure 1 biomimetics-10-00201-f001:**
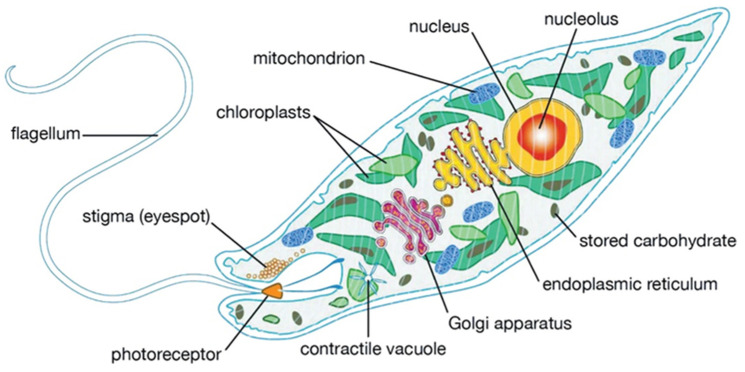
A euglena cell showing the main organelles.

**Figure 2 biomimetics-10-00201-f002:**
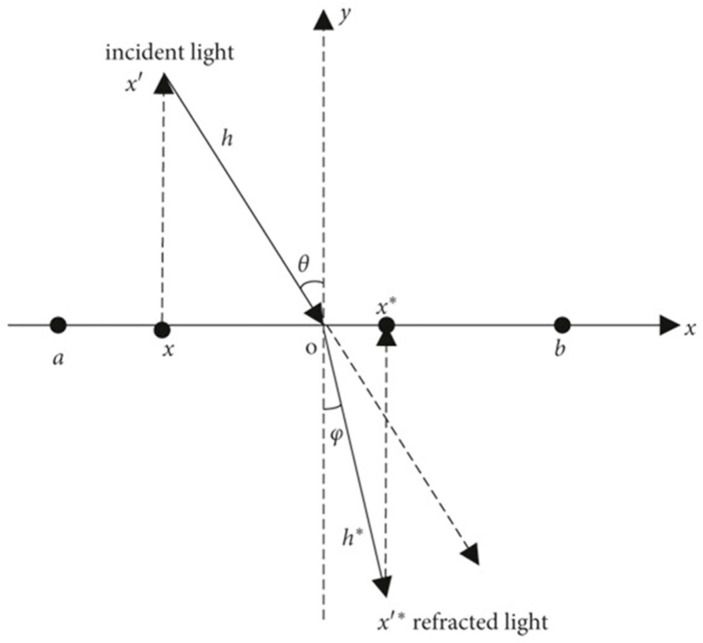
Refractive opposition-based learning model.

**Figure 3 biomimetics-10-00201-f003:**
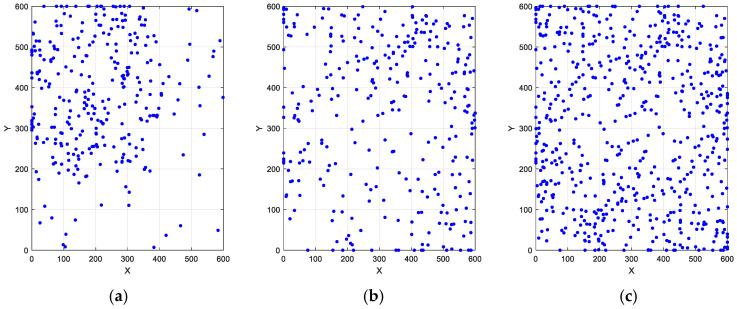
Population distributions of different schemes: (**a**) random; (**b**) tent map; (**c**) refractive opposition-based learning model.

**Figure 4 biomimetics-10-00201-f004:**
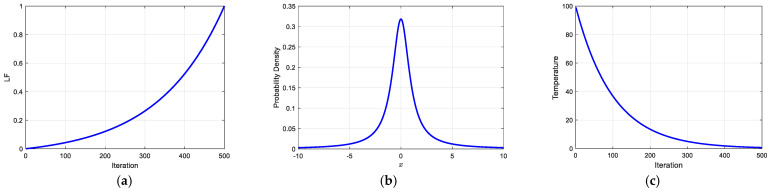
(**a**) Change curve of LF. (**b**) Standard Cauchy distribution PDF. (**c**) Temperature change curve of SA.

**Figure 5 biomimetics-10-00201-f005:**
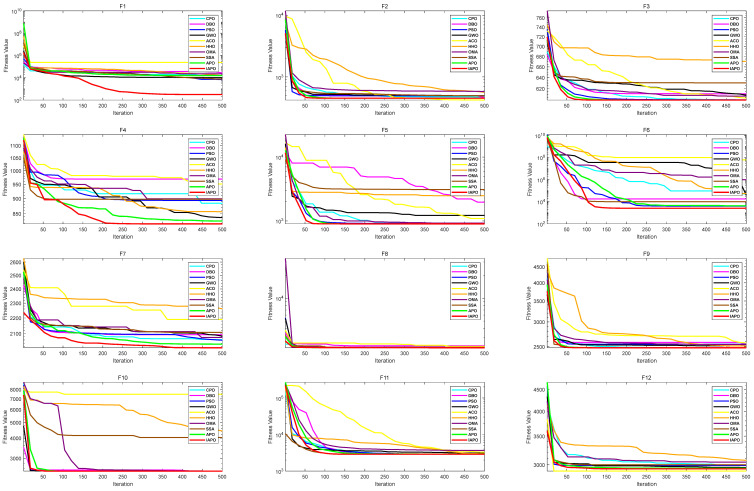
Comparison of fitness curves of 10 algorithms on CEC2022.

**Figure 6 biomimetics-10-00201-f006:**
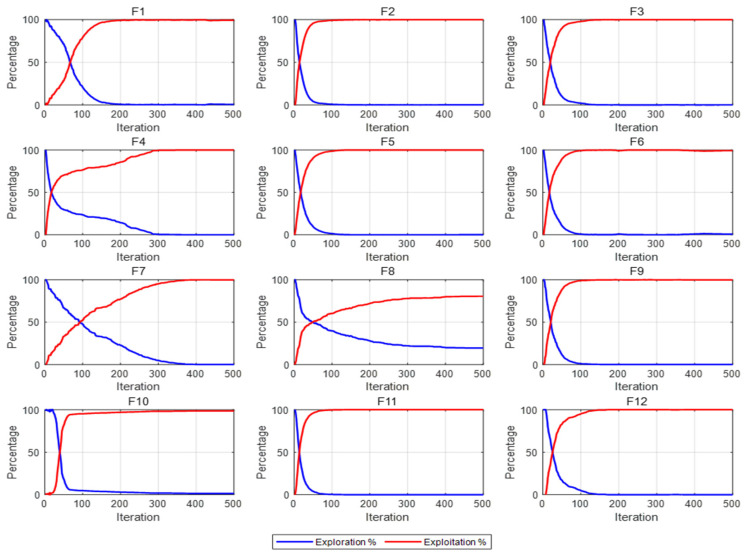
IAPO algorithm development rate and exploration rate on CEC2022.

**Figure 7 biomimetics-10-00201-f007:**
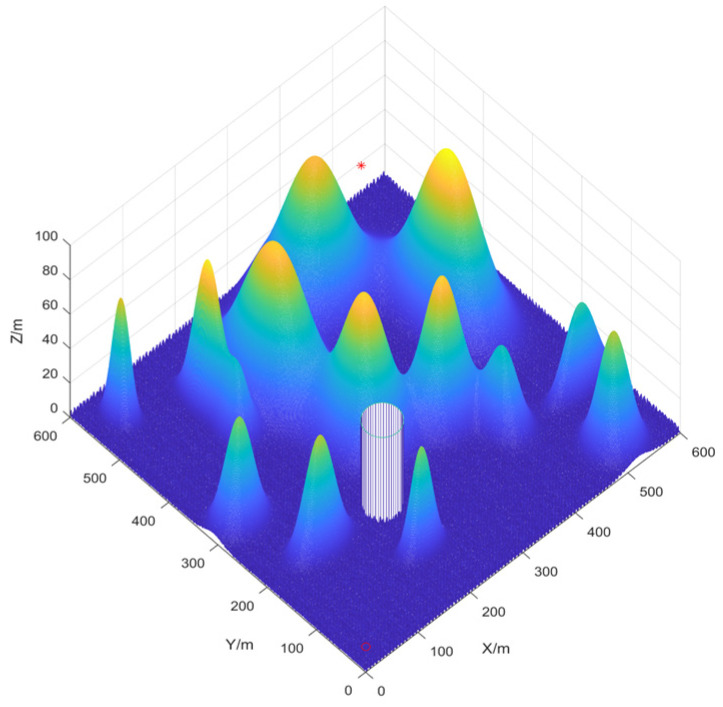
Three-dimensional mountain map.

**Figure 8 biomimetics-10-00201-f008:**
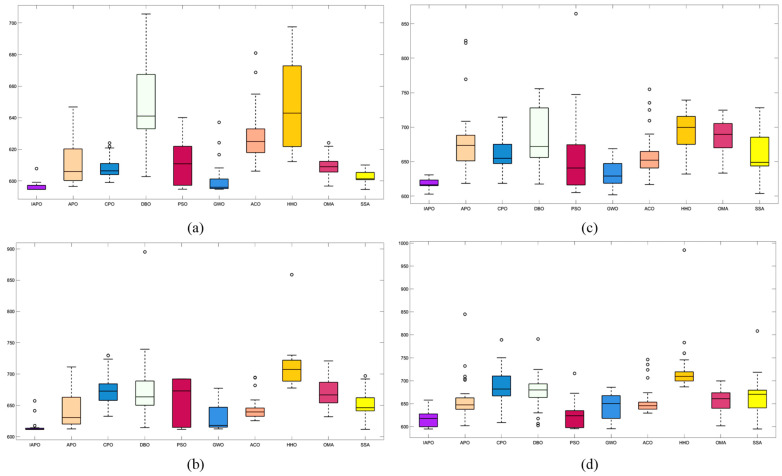
Path planning boxplots of 10 algorithms: (**a**) Map I; (**b**) Map II; (**c**) Map III; (**d**) Map IV.

**Figure 9 biomimetics-10-00201-f009:**
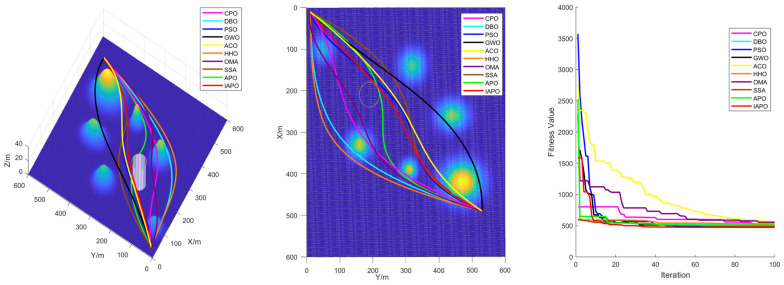
Path planning in Map I: optimal routes.

**Figure 10 biomimetics-10-00201-f010:**
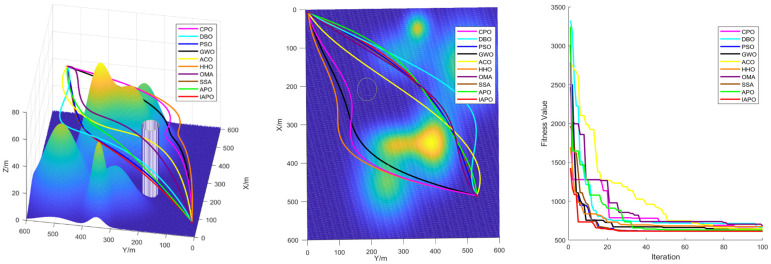
Path planning in Map II: optimal routes.

**Figure 11 biomimetics-10-00201-f011:**
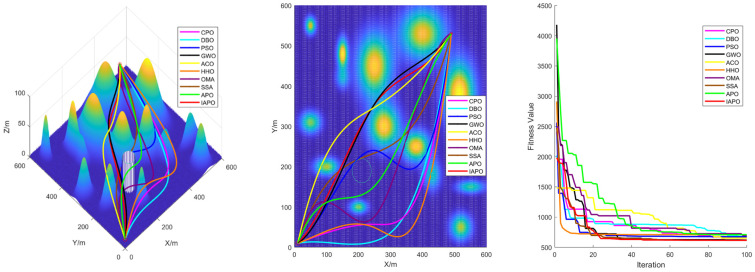
Path planning in Map III: optimal routes.

**Figure 12 biomimetics-10-00201-f012:**
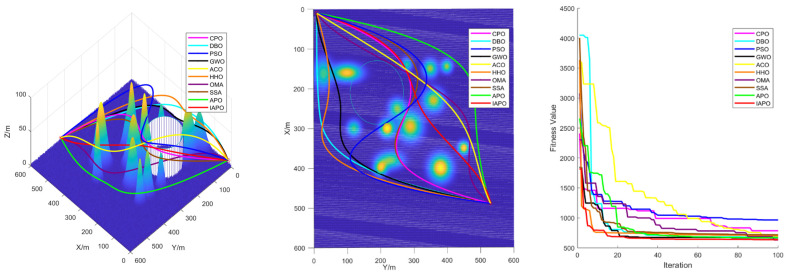
Path planning in Map IV: optimal routes.

**Table 1 biomimetics-10-00201-t001:** Benchmark functions of the 2022 IEEE Congress on Evolutionary Computation (CEC2022).

	No.	Functions	$f_{min}$
Unimodal function	F1	Shifted and full Rotated Zakharov function	300
Multimodal functions	F2	Shifted and full Rotated Rosenbrock’s function	400
	F3	Shifted and full Rotated Expanded Schaffer’s f6 function	600
	F4	Shifted and full Rotated Non-Continuous Rastrigin’s function	800
	F5	Shifted and full Rotated Levy function	900
Hybrid functions	F6	Hybrid function 1 (N = 3)	1800
	F7	Hybrid function 2 (N = 6)	2000
	F8	Hybrid function 3 (N = 5)	2200
Composition functions	F9	Composition function 1 (N = 5)	2300
	F10	Composition function 2 (N = 4)	2400
	F11	Composition function 3 (N = 5)	2600
	F12	Composition function 4 (N = 6)	2700
$\mathrm{Search}\;\mathrm{range}\;{\left[-100,100\right]}^{dim}$			

**Table 2 biomimetics-10-00201-t002:** Comparison of 10 algorithms on CEC2022.

Fun.	Index	IAPO (Ours)	APO	CPO	DBO	PSO	GWO	ACO	HHO	OMA	SSA
F1	Std	1.148 × 10^2^	3.523 × 10^3^	2.840 × 10^3^	8.522 × 10^3^	2.632 × 10^3^	4.439 × 10^3^	1.611 × 10^5^	6.307 × 10^3^	5.554 × 10^3^	4.053 × 10^3^
	Mean	4.429 × 10^2^	1.198 × 10^4^	1.075× 10^4^	2.752× 10^4^	2.943× 10^3^	1.393× 10^4^	2.651× 10^5^	1.714 × 10^4^	2.609 × 10^4^	7.182 × 10^3^
	Rank	1	5	4	9	2	6	10	7	8	3
F2	Std	1.130 × 10^1^	1.068 × 10^1^	9.966 × 10^0^	7.187× 10^1^	3.039× 10^1^	2.813× 10^1^	2.098× 10^0^	4.353× 10^1^	1.989× 10^1^	1.888× 10^1^
	Mean	4.565 × 10^2^	4.572 × 10^2^	4.579 × 10^2^	4.989 × 10^2^	4.609 × 10^2^	4.919 × 10^2^	4.202 × 10^2^	5.267 × 10^2^	4.967 × 10^2^	4.497 × 10^2^
	Rank	3	4	5	9	6	7	1	10	8	2
F3	Std	9.299 × 10^−2^	6.304× 10^−3^	8.963 × 10^−2^	1.032 × 10^1^	2.809 × 10^0^	2.664 × 10^0^	1.271 × 10^0^	6.892 × 10^0^	2.072 × 10^0^	1.348 × 10^1^
	Mean	6.000 × 10^2^	6.000 × 10^2^	6.003 × 10^2^	6.223 × 10^2^	6.031 × 10^2^	6.046 × 10^2^	6.048 × 10^2^	6.599 × 10^2^	6.085 × 10^2^	6.338 × 10^2^
	Rank	2	1	3	8	4	5	6	10	7	9
F4	Std	7.940 × 10^0^	6.106 × 10^0^	1.160 × 10^1^	2.739 × 10^1^	1.824 × 10^1^	2.728 × 10^1^	1.010 × 10^1^	1.617 × 10^1^	1.162 × 10^1^	1.583 × 10^1^
	Mean	8.227 × 10^2^	8.228 × 10^2^	8.972 × 10^2^	9.159 × 10^2^	8.529 × 10^2^	8.559 × 10^2^	9.450 × 10^2^	8.875 × 10^2^	9.104 × 10^2^	8.908 × 10^2^
	Rank	1	2	7	9	3	4	10	5	8	6
F5	Std	1.493 × 10^0^	3.503 × 10^0^	4.298 × 10^0^	5.115 × 10^2^	9.182 × 10^1^	2.102 × 10^2^	9.114 × 10^1^	2.669 × 10^2^	5.331 × 10^1^	1.569 × 10^2^
	Mean	9.012 × 10^2^	9.034 × 10^2^	9.025 × 10^2^	1.803 × 10^3^	9.418 × 10^2^	1.156 × 10^3^	1.130 × 10^3^	2.894 × 10^3^	9.508 × 10^2^	2.423 × 10^3^
	Rank	1	3	2	8	4	7	6	10	5	9
F6	Std	1.594 × 10^3^	2.309 × 10^3^	1.784 × 10^4^	6.244× 10^5^	4.909 × 10^3^	9.110 × 10^6^	7.155 × 10^7^	7.753 × 10^4^	3.647 × 10^5^	4.925 × 10^3^
	Mean	3.263 × 10^3^	4.073 × 10^3^	3.315 × 10^4^	2.029 × 10^5^	5.566 × 10^3^	3.980 × 10^6^	1.186 × 10^8^	1.428 × 10^5^	9.551 × 10^5^	6.247 × 10^3^
	Rank	1	2	5	7	3	9	10	6	8	4
F7	Std	5.309 × 10^0^	8.313 × 10^0^	8.326 × 10^0^	4.684 × 10^1^	4.237 × 10^1^	5.196 × 10^1^	3.396 × 10^1^	5.795 × 10^1^	1.295 × 10^1^	4.980 × 10^1^
	Mean	2.032 × 10^3^	2.036 × 10^3^	2.064 × 10^3^	2.110 × 10^3^	2.073 × 10^3^	2.091 × 10^3^	2.212 × 10^3^	2.180 × 10^3^	2.113 × 10^3^	2.134 × 10^3^
	Rank	1	2	3	6	4	5	10	9	7	8
F8	Std	5.520 × 10^−1^	1.342 × 10^0^	2.011 × 10^0^	7.317 × 10^1^	6.294 × 10^1^	4.799 × 10^1^	6.138 × 10^1^	1.053 × 10^2^	4.272 × 10^0^	6.800 × 10^1^
	Mean	2.223 × 10^3^	2.223 × 10^3^	2.232 × 10^3^	2.297 × 10^3^	2.262 × 10^3^	2.255 × 10^3^	2.366 × 10^3^	2.312 × 10^3^	2.242 × 10^3^	2.303 × 10^3^
	Rank	1	2	3	7	6	5	10	9	4	8
F9	Std	2.801 × 10^−3^	6.192 × 10^−1^	2.324 × 10^−1^	2.817 × 10^1^	1.694 × 10^1^	1.491 × 10^1^	3.732 × 10^1^	2.155 × 10^1^	9.034 × 10^0^	5.788 × 10^−4^
	Mean	2.481 × 10^3^	2.481 × 10^3^	2.481 × 10^3^	2.505 × 10^3^	2.491 × 10^3^	2.502 × 10^3^	2.558 × 10^3^	2.518 × 10^3^	2.513 × 10^3^	2.481 × 10^3^
	Rank	2	3	4	7	5	6	10	9	8	1
F10	Std	4.070 × 10^0^	2.079 × 10^2^	3.497 × 10^1^	8.306 × 10^2^	3.451 × 10^2^	7.538 × 10^2^	5.485 × 10^2^	7.745 × 10^2^	4.840 × 10^1^	6.223 × 10^2^
	Mean	2.502 × 10^3^	2.602 × 10^3^	2.507 × 10^3^	3.012 × 10^3^	2.919 × 10^3^	3.504 × 10^3^	7.251 × 10^3^	4.001 × 10^3^	2.519 × 10^3^	3.609 × 10^3^
	Rank	1	4	2	6	5	7	10	9	3	8
F11	Std	1.114 × 10^−1^	6.076 × 10^0^	5.137 × 10^1^	3.458 × 10^1^	1.770 × 10^0^	5.567 × 10^2^	8.876 × 10^1^	3.792 × 10^2^	1.412 × 10^2^	4.901 × 10^1^
	Mean	2.900 × 10^3^	2.911 × 10^3^	2.926 × 10^3^	2.913 × 10^3^	2.903 × 10^3^	3.560 × 10^3^	3.360 × 10^3^	3.255 × 10^3^	3.513 × 10^3^	2.937 × 10^3^
	Rank	1	3	5	4	2	10	8	7	9	6
F12	Std	4.225 × 10^0^	4.797 × 10^0^	9.071 × 10^0^	3.196 × 10^1^	3.153 × 10^1^	2.728 × 10^1^	3.869 × 10^−5^	1.449 × 10^2^	1.406 × 10^1^	2.801 × 10^1^
	Mean	2.940 × 10^3^	2.942 × 10^3^	2.990 × 10^3^	2.993 × 10^3^	2.988 × 10^3^	2.977 × 10^3^	2.900 × 10^3^	3.181 × 10^3^	3.031 × 10^3^	2.984 × 10^3^
	Rank	2	3	7	8	6	4	1	10	9	5

**Table 3 biomimetics-10-00201-t003:** Welded-beam design.

Algorithm	$x_{1}$	$x_{2}$	$x_{3}$	$x_{4}$	Optimal Cost
IAPO	0.20535	3.2388	9.036	0.20571	1.69248479
IAPO_I	0.20562	3.4759	9.0384	0.20658	1.73218539
IAPO_II	0.20586	3.3658	9.0365	0.20621	1.71439872
IAPO_III	0.20559	3.2675	9.036	0.20568	1.69651910
APO	0.20598	3.4795	9.0384	0.20671	1.73423458

**Table 4 biomimetics-10-00201-t004:** Tension/compression spring design.

Algorithm	d	D	N	Optimal Cost
IAPO	0.051302	0.355984	11.248914	0.0124131
IAPO_I	0.052247	0.369845	11.648116	0.0137789
IAPO_II	0.051764	0.356974	11.588412	0.0129975
IAPO_III	0.051845	0.359874	11.256841	0.0128234
APO	0.052351	0.369845	11.648187	0.0138339

**Table 5 biomimetics-10-00201-t005:** Path planning simulation results of 10 different algorithms in four map environments.

Map	Index	Std	Mean	Rank	Map	Index	Std	Mean	Rank
I	IAPO	2.726	596.120	1	III	IAPO	6.462	618.747	1
	APO	13.842	611.741	7		APO	48.908	681.468	7
	CPO	6.654	608.331	4		CPO	23.871	660.606	6
	DBO	22.047	643.105	9		DBO	34.700	682.779	8
	PSO	13.692	611.099	6		PSO	54.315	657.994	4
	GWO	9.683	600.348	2		GWO	18.703	633.465	2
	ACO	17.137	628.844	8		ACO	32.513	660.553	5
	HHO	28.001	647.990	10		HHO	26.528	694.809	10
	OMA	6.202	608.839	5		OMA	23.519	686.937	9
	SSA	3.529	602.106	3		SSA	34.845	657.528	3
II	IAPO	10.800	615.974	1	IV	IAPO	18.109	618.315	1
	APO	29.563	643.623	3		APO	44.660	659.520	6
	CPO	22.185	673.490	8		CPO	40.063	685.515	9
	DBO	54.947	680.091	9		DBO	37.149	679.394	8
	PSO	34.967	657.843	6		PSO	28.245	626.163	2
	GWO	20.417	632.021	2		GWO	30.816	641.589	3
	ACO	17.650	643.988	4		ACO	30.870	656.397	4
	HHO	32.769	709.538	10		HHO	53.868	721.477	10
	OMA	20.673	670.385	7		OMA	23.622	658.574	5
	SSA	25.322	648.945	5		SSA	40.118	662.862	7

## Data Availability

Data are contained within the article.

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
