# Peer review of "Three-Dimensional UAV Path Planning Based on Multi-Strategy Integrated Artificial Protozoa Optimizer"

_biomimetics, 2025, doi:10.3390/biomimetics10040201_

Round 1

Reviewer 1 Report

Comments and Suggestions for Authors

The paper describes a method for UAV path planning as an improvement of APO algorithm. It is well written and fits the journal scope.

In my opinion, some aspects should be addressed in order to proceed to publication.

The APO equations in section 2.1 have missing variable explanations.

Figure 3 3D representation does not contribute specially. 2D proyections or histograms could be more informative, or it can simply be removed if no relevant analysis is derived from it.

F8 function in explor/explot evolution differs from others in not reaching 0/100% limits. What's the reason for this ?

What values for weights w1, w2 and w3 have been considered ? How are the paper results affected by changes in those values ?

Aren't the no-fly zones too small to represent any difference compared to peaks ?

Path quality is hard to judge from 3D representations. 2D proyections and speed/acceleration profiles should be included.

Does H factor properly capture the high cost of small but frequent altitude changes around average value ? Again, a detailed altitude profile could also be informative here.

I consider the warfare reference unnecessary.

Other minor issues:

missing . in kn multiplication

"fractionpf"

"improvements significantly improved"

Using 'Cauchy' name in the equation is confusing, consider an abreviation instead.

end if missing before line 42 in algorithm 1

Best values by function should be bold marked in table 2

Reviewer 2 Report

Comments and Suggestions for Authors

The manuscript introduces a novel multi-strategy integrated Artificial Protozoa Optimizer (IAPO) tailored for three-dimensional UAV path planning. By enhancing the traditional APO through the integration of a Tent Map and Refractive Opposition-Based Learning for population initialization, a dynamic optimal leadership mechanism with a nonlinear adjustment factor, a Cauchy mutation strategy, and simulated annealing, the proposed method aims to overcome common challenges such as slow convergence and local optima entrapment. Extensive experiments on CEC2022 benchmark functions, alongside comparisons with nine state-of-the-art algorithms, demonstrate IAPO's superior performance in convergence speed, solution accuracy, and robustness, as well as its effectiveness in generating collision-free, energy-efficient UAV trajectories in complex 3D environments. I have some concerns:

  1. The manuscript fails to clearly justify the rationale behind selecting the multi-strategy integrated Artificial Protozoa Optimizer (IAPO). While the proposed modifications (Tent Map, ROBL, dynamic leadership, etc.) are described, the authors do not convincingly explain why these specific strategies are expected to outperform alternative methods.
  2. There is insufficient discussion on the advantages of the chosen method in relation to the specific UAV path planning problem. For example, while improved convergence is claimed, the connection between the algorithm’s design choices and the real-world challenges of dynamic obstacle avoidance or energy minimization is underdeveloped. Specific examples from the text (e.g., lines describing the leadership mechanism and its role in guiding the search) should be used to illustrate how these improvements overcome known limitations in conventional bio-inspired heuristics.
  3. The Introduction mixes background, motivations, and contributions in a single narrative. It would be beneficial to divide the section into distinct subsections—for instance, “Motivations” and “Contributions”—to clearly articulate the problem’s significance, the gap in the existing literature, and the novel aspects of the work. A more focused review of the literature, explicitly contrasting IAPO with competing approaches, would help readers appreciate the innovation and necessity of the proposed method.
  4.  Equation (37), which aggregates the costs associated with path length, height, and curvature, must be normalized. Without normalization, differences in the scales of these individual components could disproportionately influence the optimization process. For instance, if the path length is measured in hundreds of meters while curvature is a unitless parameter varying in a smaller range, the weight factors (w₁, w₂, w₃) might not effectively balance the contributions of each term. Normalizing each term ensures that all components contribute comparably to the total cost, thereby enabling a more balanced optimization. This adjustment would help prevent any single aspect—such as path length—from dominating the objective function, leading to a more robust and reliable UAV path planning solution.
  5. The manuscript presents the basic framework for constraint handling through Equations (38) and (39), which define the initial conditions and collision detection for UAV path planning. However, the overall strategy for enforcing constraints within the optimization process remains unclear. It is not explicitly stated whether constraint violations are managed by applying penalty functions, repairing infeasible solutions, or through another mechanism. A clearer explanation of how these constraints are integrated into the optimization algorithm would help readers understand how the method ensures feasibility and balances the trade-off between exploring new solutions and adhering to practical limitations in complex environments.
  6. The experimental setup, though extensive, lacks clarity in some key areas. The rationale behind the parameter settings (e.g., population size, iteration limits, decay coefficients) is not thoroughly justified, and the sensitivity of the results to these choices is not analyzed.
  7. The manuscript would benefit from the inclusion of sensitivity analyses and ablation studies to isolate the contributions of each component (e.g., ROBL, Cauchy mutation, SA integration). Such analyses would strengthen the argument that every component is necessary and effective.
  8. While comparisons with state-of-the-art methods are presented, the paper does not sufficiently detail how the benchmarks and evaluation metrics were selected. It is recommended that further comparisons be made—not only with more algorithms but also by testing on real-world UAV path planning scenarios—to validate the practical relevance of the approach.
  9. Additionally, the consistency and clarity of acronyms (both in the title and at first mention) need to be revisited. Some terms (e.g., “APO” vs. “IAPO”) could be confusing if not defined clearly and used consistently throughout the manuscript.
  10. The limitations of the proposed approach (e.g., computational complexity or challenges in real-time applications) are not adequately discussed. Explicitly addressing these issues would provide a more balanced view and suggest directions for future work.
  11. The conclusion, while summarizing the experimental results, remains overly concise and does not capture the broader impact of the research. A more articulate conclusion should synthesize key findings and discuss potential managerial or operational implications—particularly how the method could be applied in different domains (e.g., logistics, disaster management).
  12. The logical flow between sections can be improved. For instance, transitions between the method description and experimental analysis appear abrupt, and the pseudocode could be better integrated into the narrative to help readers follow the algorithm’s progression. Some mathematical formulations (e.g., in the sections on the Tent Map and ROBL) would be clearer if additional explanatory notes or examples were provided.
  13. It is advised that the authors include a dedicated section that critically compares IAPO’s performance with existing literature not only in numerical benchmarks but also in qualitative aspects such as robustness to environmental changes.
  14. Further clarification on the theoretical underpinnings of the integrated strategies (e.g., how the dynamic optimal leadership mechanism adjusts exploration–exploitation balance in a quantifiable manner) would add depth to the analysis.
  15. The discussion regarding the pseudocode is overly condensed; more detailed commentary on the algorithm’s decision logic and convergence properties would aid reproducibility.
  16. Finally, the abstract and introduction should be reworked to clearly state the novelty of the approach, with explicit statements on how the method addresses the identified gaps in literature.

Overall, while the manuscript presents an interesting multi-strategy approach to UAV path planning, the current presentation suffers from a lack of detailed justification and in-depth analysis. Significant revisions focused on clarity, thoroughness of experimental validation, and critical comparison with existing methods are needed before further consideration.

Comments on the Quality of English Language

The manuscript would benefit from a thorough language edit. Several sections contain verbose sentences and complex expressions that obscure the technical content. Improving clarity and coherence would enhance readability.

Round 2

Reviewer 1 Report

Comments and Suggestions for Authors

The review questions have been properly addressed.

However, I have a concern about the algorithm comparison presented in Table 2. Comparison of 10 algorithms on CEC2022. What versions of the competitors algorithms have been implemented ? After some verification, the optimization values seem not to match the corresponding ones published by the contestants in that competition, so I understand a basic not fine tuned algorithm has been tested.

In order to evaluate the quality of the proposal, the tuning process applied for the compared algorithms and the authors proposal should be explained.

Reviewer 2 Report

Comments and Suggestions for Authors

All my comments from the last round have been satisfactorily addressed, and the manuscript is now publishable.
